# A VLP-Based Vaccine Displaying HBHA and MTP Antigens of *Mycobacterium tuberculosis* Induces Protective Immune Responses in *M. tuberculosis* H37Ra Infected Mice

**DOI:** 10.3390/vaccines11050941

**Published:** 2023-05-04

**Authors:** Juan Wang, Tao Xie, Inayat Ullah, Youjun Mi, Xiaoping Li, Yang Gong, Pu He, Yuqi Liu, Fei Li, Jixi Li, Zengjun Lu, Bingdong Zhu

**Affiliations:** 1Gansu Provincial Key Laboratory of Evidence Based Medicine and Clinical Translation, Lanzhou Center for Tuberculosis Research, Institute of Pathogen Biology, School of Basic Medical Sciences, Lanzhou University, Lanzhou 730000, China; wangj19@lzu.edu.cn (J.W.); xiet18@lzu.edu.cn (T.X.); miyj@lzu.edu.cn (Y.M.); gongy20@lzu.edu.cn (Y.G.); lf@lzu.edu.cn (F.L.); 2Institute of Pathogenic Physiology, School of Basic Medical Sciences, Lanzhou University, Lanzhou 730000, China; 3Respiratory Department of Lanzhou Pulmonary Hospital, Lanzhou 730000, China; 4State Key Laboratory of Genetic Engineering, School of Life Sciences, Fudan University, Shanghai 200438, China; lijixi@fudan.edu.cn; 5Lanzhou Veterinary Research Institute, Chinese Academy of Agricultural Sciences, Lanzhou 730000, China; 6State Key Laboratory for Animal Disease Control and Prevention, College of Veterinary Medicine, Lanzhou University, Lanzhou 730000, China

**Keywords:** *Mycobacterium tuberculosis*, virus-like particles, subunit vaccine, Heparin-binding hemagglutinin, *Mycobacterium tuberculosis* pili

## Abstract

Heparin-binding hemagglutinin (HBHA) and *M. tuberculosis* pili (MTP) are important antigens on the surface of *Mycobacterium tuberculosis*. To display these antigens effectively, the fusion protein HBHA-MTP with a molecular weight of 20 kD (L20) was inserted into the receptor-binding hemagglutinin (HA) fragment of influenza virus and was expressed along with matrix protein M1 in Sf9 insect cells to generate influenza virus-like particles (LV20 in short). The results showed that the insertion of L20 into the envelope of the influenza virus did not affect the self-assembly and morphology of LV20 VLPs. The expression of L20 was successfully verified by transmission electron microscopy. Importantly, it did not interfere with the immunogenicity reactivity of LV20 VLPs. We demonstrated that LV20 combined with the adjuvant composed of DDA and Poly I: C (DP) elicited significantly higher antigen-specific antibodies and CD4^+^/CD8^+^ T cell responses than PBS and BCG vaccination in mice. It suggests that the insect cell expression system is an excellent protein production system, and LV20 VLPs could be a novel tuberculosis vaccine candidate for further evaluation.

## 1. Introduction

*Mycobacterium tuberculosis (M. tuberculosis)*, an ancient intracellular bacterial pathogen, is still causing around ten million new cases and 1.6 million deaths worldwide in 2021 [1]. Although *Mycobacterium bovis* bacillus Calmette–Guérin (BCG), the only available vaccine used in clinics, provides effective protection against severe TB in childhood, it fails to provide consistent protection for adults [2,3] and can cause fatal infections in immunocompromised hosts [4]. Therefore, it is urgent to develop novel vaccines to prevent TB in adults [5]. The recombinant subunit protein vaccines are considered generally safe and well tolerated. In a phase IIB clinical trial, M72/AS01E, being composed of Mtb32A and Mtb39A antigens in adjuvant AS01E, could prevent latent TB from progression in adults for at least 3 years with an efficacy of 49.7% [6,7]. It gives hope that subunit vaccines have the potential to induce protection against TB in adults.

Both cellular and humoral immunity provides protection against tuberculosis. Existing studies have shown that T cell-mediated immune responses provide protective immunity against *M. tuberculosis* infection [8,9]. Therefore, most of the current TB vaccine candidates, such as ID93/GLA-SE [10], H56:IC31 [11], and GamTBVac [12], contain secreted proteins, which are supposed to induce cell-mediated immunity (CMI). Besides CMI, increasing evidence suggests that antibodies also play an active role in protecting against *M. tuberculosis* [13,14]. The monoclonal antibodies against surface proteins and polysaccharides of mycobacteria have been shown to enhance protective host immune responses [15,16,17,18,19]. For example, the mycobacterial lipoarabinomannan-specific monoclonal antibody prolonged the survival of mice infected with *M. tuberculosis* [16]. Passive inoculation with IgA monoclonal antibody against the α-crystallin antigen of *M. tuberculosis* reduced the bacterial load in the lungs of infected mice [19].

The *M. tuberculosis* cell envelope proteins play a critical role in *M. tuberculosis* resistance and survival and are supposed to be the primary target for developing new vaccines. Among the cell wall proteins of mycobacteria, HBHA plays an important role in the extrapulmonary dissemination of *M. tuberculosis* by binding to non-phagocytic cells [20]. Disruption of the HBHA gene significantly affects mycobacterial interactions with epithelial cells [21]. BCG coated with anti-HBHA antibodies significantly reduced spleen colonization compared to uncoated BCG, suggesting that antibody responses to HBHA may enhance immune protection against tuberculosis [20]. Moreover, HBHA can manipulate actin polymerization and depolymerization processes, allowing *M. tuberculosis* to escape from the phagosome into the cytosol, which contributes to latent tuberculosis infection [22,23]. Unlike most patients with active pulmonary tuberculosis, individuals with latent tuberculosis infection produce greatly increased amounts of IFN-γ in response to HBHA [24]. HBHA-specific CD8^+^ T lymphocyte-mediated protective immune response of individuals with latent TB supports the notion that HBHA may be protective against bacterial clearance in humans [25].

Besides HBHA, *M. tuberculosis* pili (MTP), encoded by Rv3312A, is another important cell wall-associated protein. It mediates the adhesion of *M. tuberculosis* to host cells and contributes to biofilm formatting. *M. tuberculosis* strains lacking MTP showed reduced ability to adhere and invade A549 cells (lung epithelial cells) by 69.39% (*p* = 0.047) and 56.20% (*p* = 0.033) compared with wild-type strains [26]. The complemented strain restored their adhesion and invasion to wild-type levels. In addition, biofilm formation by the Δmtp mutant was reduced by 68.4% compared with the wild-type strain [27]. Existing scientific research shows that MTP is considered to be a virulence factor and, thus, an important target for vaccine development.

Virus-like particles (VLPs) constitute a safe and valuable platform in clinical vaccine development. VLPs have many advantages over traditional subunit vaccines. It can exhibit heterologous antigens from different pathogens on the surface of VLPs, mimic the size and shape of viruses, and enhance the probability of interaction with antigen-presenting cells and B cells. Although many VLPs have adjuvant-like effects properties, VLP-based vaccines without adjuvants demonstrate limited immunogenicity, and using adjuvants in VLPs formulations may increase the antigen-specific immune response [28]. The most well-known adjuvant, aluminum salt, has been used in all approved VLPs vaccines. In addition, the lack of a viral genome in VLPs provides a better safety profile for immunocompromised and elderly individuals. VLPs can be produced in several different expression systems, including bacteria, insect, yeast, or mammalian cells, and plant expression systems [29,30,31,32]. Baculovirus-insect (B/IC) cell expression systems remain the most promising method for vaccine production. Using the B/IC expression system, a number of potential VLP-based vaccines for human diseases have been developed, some of which have been approved or are in clinical trials, such as Cervarix^®^ for Human Papilloma Virus, NanoFlu™ for Seasonal Influenza, ResVax™ for Respiratory Syncytial Virus, and Ebola GP Vaccine for Ebola [33]. Previously, we utilized the B/IC expression system to display protective epitopes of SARS-CoV-2, and verified that SARS-CoV-2 VLPs were successfully self-assembled in insect cells after infection [34]. Here, we explored whether the B/IC expression systems could be used to display the surface antigens of M. tuberculosis along with matrix protein M1 of the influenza virus, and whether the insertion of heterologous antigens could affect the folding and assembly of the virus-like particles. VLPs LV20 were identified by transmission electron microscopy. The antigen-specific immune response and protective efficacy against H37Ra were evaluated in mouse models.

## 2. Materials and Methods

### 2.1. Bioinformatic Analysis

The hydrophobicity and stability of HBHA, MTP, and the fusion protein LT20 were predicted by the ExPasy ProtParam tool (http://web.expasy.org/protparam/; accessed on 5 October 2022). The outer membrane portions of HBHA and MTP with hydrophilic properties were selected for fusion protein L20. T cell and B cell epitopes of HBHA and MTP were predicted by the Immune Epitope Database (IEDB; http://www.iedb.org/; accessed on 24 October 2022). The 3-D structures of HBHA, MTP, and fusion protein L20 were predicted by AlphaFold2 (https://wemol.wecomput.com; accessed on 24 December 2022).

### 2.2. Animals

Specific pathogen-free female C57BL/6 mice (6–8 weeks old) were purchased from Lanzhou Veterinary Research Institute, Chinese Academy of Agricultural Sciences (Lanzhou, China). All animals received free access to water and standard mouse chow. For immunodetection experiments, mice were housed in special pathogen-free environment at Gansu University of Traditional Chinese Medicine. For *M. tuberculosis* H37Ra challenge experiments, animals were maintained in P2 laboratory at Lanzhou University, China. All animal experiments were carried out under the guidelines of Council on Animal Care and Use, and the protocols were reviewed and approved by Institutional Animal Care and Use Committee of Lanzhou University.

### 2.3. Cell Lines and Cell Culture

ExpiSf9™ cells (Gibco™, Waltham, MA, USA) were maintained in suspension culture in serum-free ExpiSf9™ CD medium (Gibco™, Waltham, MA, USA) at 27 °C with shaking at 125 rpm. Cell numbers were counted under a microscope to determine cell density, and trypan blue stain exclusion was used to determine cell viability.

### 2.4. Construction of LV20 VLPs Presenting HBHA and MTP

#### 2.4.1. Construction of pFastBacDual-MH-L20 Recombinant Double Expression Plasmid

The pFastBacDual vector has two promoters, pP10 and pPH. The transmembrane region and extracellular signal peptide of HA and the matrix protein M1 of influenza virus were inserted downstream of the pP10 and pPH promoters, respectively (Figure 1A). The restriction endonuclease recognition sites for *Nde I* and *EcoR I* were designed between the extracellular signal peptide and the transmembrane region of HA, and the plasmid was named pFastBacDual-MH plasmid. The DNA sequence coding fusion protein of HBHA_109–199_-MTP_27–89_ containing the *Nde I* and *EcoR I* was generated by PCR amplification from a template plasmid pET30 -HBHA_109–199_-MTP_27–103_ constructed previously in our laboratory. The universal upstream primer was TAATACGACTCATATAGGGGAATTG, and the universal downstream primer was GGAATTCATCATCCAGCACCGGGCCTTC. Furthermore, the gene encoding the C-terminal domain of HBHA_109–199_ and the outer membrane region of MTP_27–89_ were inserted into the restriction sites of the HA fragment to construct a recombinant double expression plasmid pFastBacDual-MH-L20 (Figure 1A).

#### 2.4.2. Construction of Recombinant Baculoviruses

Recombinant baculoviruses were generated using the Bac-to-Bac expression system (Invitrogen, Carlsbad, CA, USA). Briefly, the recombinant double expression plasmid pFastBacDual-HM-L20 was transformed into DH10 Bac™ *E. coli* (Invitrogen, Carlsbad, CA, USA) strain after two screening tests of blue and white dots; white monoclonal colonies that contained the recombinant bacmid -HM-L20, were selected and verified by PCR. ExpiSf9™ cells grown to a cell density of approximately 5 × 10^6^ − 10 × 10^6^ viable cells/mL (90% viable cells) and 62.5 × 10^6^ cells were transfected with 2.5 µg of bacmid -HM-L20. Transfected ExpiSf9™ cells were grown at 27 °C and 125 rpm. When the cytopathic index reached 30%, the cell supernatant was collected to obtain recombinant baculovirus (Figure 1B).

#### 2.4.3. Production and Purification of LV20 VLPs

ExpiSf9™ cells were infected with the recombinant baculovirus at the rate of 50:1. Supernatants were collected on day 4 by centrifugation at 3000× *g* for 5 min, and the pellet was discarded after centrifugation at 8000× *g* for 30 min. Finally, the VLPs displaying L20 (LV20 in short) were centrifuged through a 20%–35%–60% discontinuous sucrose gradient at 100,000× *g* for 1 h at 4 °C. The white bands between 35% and 60% were collected and diluted with PBS, and the pellet was collected by centrifugation at 100,000× *g* for 1 h at 4 °C. The samples were diluted with PBS and stored at −80 °C for the following analysis.

### 2.5. Western Blot Analysis

The samples were characterized by Western blot (WB) analysis. LV20 VLPs samples were subjected to 15% SDS-PAGE and transferred to polyvinylidene difluoride (PVDF) membrane by a transfer device. Western Blot analysis was performed using anti-M1 monoclonal rabbit antibody or anti-MTP polyclonal mouse antibody as primary antibody. Alkaline phosphatase-conjugated anti-rabbit or anti-mouse IgG (ImmunoWay, Plano, TX, USA) was used as the secondary antibody. Proteins were then stained using a chromogenic kit (Promega Corporation, Madison, WI, USA). The results were analyzed using a chemiluminescence imaging system (Vilber Lourmat, Marne-la-Vallée, France).

### 2.6. Transmission Electron Microscopy

#### 2.6.1. The VLPs Were Visualized by Transmission Electron Microscopy (TEM)

Exposing the copper mesh to UV light for 10 min. A drop of the VLP sample was deposited onto the carbon-coated copper grid. After two minutes, the extra sample was taken out using filter paper burrs. The samples were then stained with 1% phosphotungstic acid for 30 to 60 s. Filter paper burrs were used to remove the staining solution, and the sample was left to dry for 30 min at room temperature. The VLPs were detected by TEM. Observe the samples at 200 kV and 100–200 k magnification by the FEI Talos F200C TEM (FEI, Brno, Czech Republic).

#### 2.6.2. Detection of HBHA-Specific Gold Nanoparticles on the Surface of VLPs by TEM

After a drop of VLP sample was deposited onto the carbon-coated copper grid, the sample was blocked at room temperature with 5% BSA for 1 h. After three rounds of washing in PBS, anti-mouse HBHA polyclonal serum was applied dropwise (1:400) and incubated for two hours at room temperature. The copper grid was washed three times before 5 uL of diluted anti-mouse colloidal gold (BOSTER) was applied. The sample was then incubated at room temperature for an hour. Finally, the sample was stained with 1% phosphotungstic acid for 30 to 60 s. Filter paper burrs were used to remove the staining solution, and the sample was left to dry for 30 min at room temperature. The aggregation of gold nanoparticles on the surface of virus-like particles was observed using TEM. The samples were observed at 200 kV and 100–200 k magnification by the FEI Talos F200C TEM (FEI, Brno, Czech Republic).

### 2.7. Vaccine Preparation and Immunization

BCG (Danish strain) bacteria and *M. tuberculosis* H37Ra strain (ATCC25177) were cultured in Sauton medium. Single mycobacterial proteins HBHA with His tag were purified by Ni-NTA His column (Novagen, Madison, WI, USA). The mice were vaccinated subcutaneously with BCG (5 × 10^5^ CFU in 100 μL per mouse). The protein HBHA (10 μg/dose) and virus-like particles LV20 (50 μg/dose) were mixed with an adjuvant composed of dimethyl dioctadecylammonium (DDA) (Anhui Super chemical technology Co, Ltd., Anhui, China) (250 μg/dose) and Poly I: C (Kaiping Genuine Biochemical Pharmaceutical Co, Ltd., Guangdong, China) (50 μg/dose) and LV20 (50 μg/dose) without adjuvant as vaccines. To observe the immune efficacy, we immunized C57BL/6 mice with BCG and PBS one-time inoculation at day 0. HBHA/DP, LV20, and LV20/DP were injected subcutaneously three times on days 0, 14, and 28 (Figure 2). Flow cytometry and intracellular cytokine staining (ICS) and serum antibody profiles were evaluated on day 70. The number of mice per group was at least 3.

### 2.8. Flow Cytometry for Intracellular Cytokine Analysis

Forty-two days after the last immunization, lymphocytes were isolated from spleens and stimulated with HBHA (5 μg/mL) in vitro for 4 h. Protein transport inhibitor cocktails (eBioscience, San Diego, CA, USA) were added to each well and incubated at 37 °C under 5% CO_2_ for 6–7 h. Samples were stained with anti-CD4-FITC and anti-CD8-PerCP-Cy5.5, then permeabilized with BD Cytofix/Cytoperm and stained with IFN-γ-APC and anti-IL-2-PE staining kits according to the manufacturer’s instructions. All samples were analyzed using ACEA NoveCyte. Results were analyzed and calculated byNovoCyte flow cytometer (ACEA Biosciences, San Diego, CA, USA). All Abs were purchased from BioLegeng (San Diego, CA, USA) unless stated. Flow cytometry gating strategy is shown in Appendix A. The spleen lymphocytes were gated first on parameters SSC-H and FSC-H (lymphocytes), then single cells were selected by the parameters FSC-H and FSC-A (single cells). Frequencies of HBHA-specific CD4^+^ IFN-γ^+^ T cells, CD4^+^ IL-2^+^ T cells, CD8^+^ IFN-γ^+^ T cells, and CD8^+^ IL-2^+^ T cells were analyzed using flow cytometry. Bar graphs show the percentages of cytokine-producing T cells.

### 2.9. Enzyme-Linked Immunosorbent Assay (ELISA) for Antigen-Specific Antibodies in Mouse Sera

Mice serum was collected on day 70 to determine immunoglobulin IgG, IgG1, and IgG2c antibody titers by ELISA. HBHA antigens (0.5 μg/100 μL) were coated on 96-well ELISA microtiter plates at 4 °C overnight. A solution of 5% BSA was used to block non-specific binding sites at 37 °C for 1 h. Diluted serum samples (from 1:100 to 1:12,800) were added to each well and incubated at 37 °C for 2 h. Horseradish peroxidase-conjugated goat anti-rabbit IgG, IgG1, and IgG2c (Bersee, Beijing, China) were used at 1:5000 to detect each isotype. After complete washing and 15 min of incubation with the TMB substrate solution, stop solution was added, and optical density at 450 nm was determined using a microplate reader.

### 2.10. M. tuberculosis Challenge and Bacteria-Load Detection

C57BL/6 mice were immunized subcutaneously with LV20, HBHA/DP, and LV20/DP on days 0, 28, and 84 with PBS and BCG as controls. Eighty-four days after the last immunization, the immunized mice were anesthetized with 4% chloral hydrate (0.1 mL/10g) and challenged with 5 × 10^6^ CFU of *M. tuberculosis* H37Ra via the aerosol route. By plating serial dilutions of lung homogenates on 7H10 agar plates and counting colony-forming units following incubation at 37 °C, bacterial counts per mg of lung tissue were determined at 21 days after infection.

### 2.11. Statistical Analysis

The experimental data were expressed as Mean ± SD. The data were analyzed by GraphPad Prism 5.0 software with conducted two-tailed Student′s *t*-tests to compare two groups. One-way ANOVA analysis followed by Tukey post hoc test was used to compare multiple groups. *p* < 0.05 was considered statistically significant.

## 3. Results

### 3.1. Bioinformatic Analysis of HBHA and MTP

Since the outer membrane portion of bacterial cell wall protein is the potential fragment to interact with host cells, the outer membrane portion with high immunogenicity of HBHA and MTP was chosen to construct the subunit vaccine. The C-terminal domain of HBHA (from 109 to 199) and the outer membrane portion MTP (from 27 to 89) were selected to construct the fusion protein L20 (Table 1 and Figure 1A). T and B cell epitopes of HBHA and MTP were predicted by IEDB, which suggests that selected sequences were rich in T cell and B cell epitopes (Table 2). The hydrophobicity and stability of fusion protein were obtained by the ProtParam Tool. The instability index of HBHA (46.67) and MTP (54.14) demonstrated that these two proteins were unstable (II > 40 indicates instability). The instability index of L20 was 36.2, classifying the fusion protein as stable. The Grand average of hydropathicity (GRAVY) of HBHA, MTP, and fusion protein L20 was very low, indicating their high affinity for water. The 3-D structures of HBHA (Figure 3A), MTP (Figure 3B), and L20 (Figure 3C) were predicted by AlphaFold2. The chosen sequences for L20 were represented in red (HBHA_100–199_) and purple (MTP_27–89_) color, respectively (Figure 3C). The results showed that the natural structure of selected portions of HBHA and MTP was kept in fusion protein L20, which indicated that L20 might have the immunogenicity of HBHA and MTP to some degree.

### 3.2. Construction of Recombinant LV20 VLPs

HBHA has three functional regions: a transmembrane region composed of 18 amino acids near the N-terminus; an α-helical structure composed of 81 amino acids; and a C-terminal rich in lysine, proline, and alanine [14] (Figure 1A). The interaction between HBHA and host components is mainly mediated by the C-terminal domain [4,35]. The MTP consists of an inner membrane region, a transmembrane region, and an outer membrane region (Figure 1A). The outer membrane region of MTP is rich in B cell and T cell epitopes (Table 2). The gene sequences encoding the C-terminal domain of HBHA_109–199_ and the outer membrane region of MTP_27–89_ (L20) were inserted between the transmembrane region and extracellular signal peptide of the influenza virus HA gene to construct a recombinant double expression plasmid pFastBacDual-MH-L20 (Figure 1A). The recombinant double expression plasmid was transformed into DH10 Bac™ *E. coli* strain, and positive clones containing the recombinant bacmid -HM-L20 were obtained by antibiotic and blue–white screening. The correct recombinant bacmid DNA was extracted and transfected into Sf9 cells to obtain recombinant baculovirus (Figure 1B). The baculovirus was transfected into ExpiSf9 cells. At last, L20 and MH were expressed and self-assembled to form the influenza virus VLPs containing L20 (LV20 for short; Figure 1B).

### 3.3. Identification of LV20 VLPs by Transmission Electron Microscopy and Western Blotting

The morphology of LV20 VLPs was observed by transmission electron microscopy. LV20 VLPs exhibited spherical structures with diameters ranging from 80 to 120 nm (Figure 4A). The results indicate that LV20 VLPs can be expressed in infected insect cells and self-assemble in a morphology similar to the influenza virus. To confirm the displaying of HBHA protein in VLPs, immunogold-labeled LV20 VLPs were analyzed by transmission electron. Using HBHA polyclonal antibody as the primary antibody, gold particles were detected on the surface of LV20 VLPs (Figure 4B), indicating that HBHA was displaying on the outer surface of VLPs. To confirm the displaying of MTP and M1 in the virus-like particles, LV20 VLPs were verified by WB using MTP protein polyclonal antibody and M1 protein monoclonal antibody as primary antibodies. The fusion protein containing L20 and the transmembrane region and extracellular signal peptide of the influenza virus HA were approximately 27 kDa (Figure 4C, Lane 4), and the M1 of 29 kDa (Figure 4C, Lane 1) is consistent with the predicted molecular weight. The result showed that LV20 VLPs contained the MTP protein (Figure 4C, Lane 4). In addition, the fusion antigen LT28 (Figure 4C, Lane 3) containing MPT64, Rv1978, and Rv2645 of *Mycobacterium tuberculosis* antigens is independent of LV20 and can be used as a negative control, while the Pili (Figure 4C, Lane 2) antigen can be used as a positive control.

### 3.4. Cell-Mediated Immune Responses Induced by Vaccination with LV20

HBHA/DP, LV20, and LV20/DP were injected subcutaneously on days 0, 14, and 28. (Figure 2). Twenty-one days after the last immunization, the intracellular IFN-γ and IL-2 produced by spleen CD4^+^ and CD8^+^ T lymphocytes following HBHA (5 ug/mL) stimulation were analyzed by flow cytometry. The frequencies of IFN-γ secreting CD4^+^ and CD8^+^ T lymphocytes from the group of LV20 were higher than those of the HBHA/DP and BCG groups (*p* < 0.05; Figure 5A,B). The frequencies of IL-2 secreting CD4^+^ and CD8^+^ T lymphocytes from the group of LV20 were similar to the HBHA/DP group but higher than the BCG group (*p* < 0.05; Figure 4A,B). Furthermore, the LV20/DP group produced higher frequencies of IFN-γ- and IL-2-producing CD4^+^ and CD8^+^ T lymphocytes than the HBHA/DP and BCG groups (*p* < 0.05; Figure 5A,B). The results indicated that LV20 vaccination promoted stronger responses than the individual HBHA antigen. Furthermore, compared with the LV20 group, the LV20/DP group produced higher frequencies of IFN-γ and IL-2-producing CD4^+^ and IFN-γ-producing CD8^+^ T lymphocytes (*p* < 0.05; Figure 5A,B), which suggests that DDA combined with Poly I:C is an effective vaccine adjuvant to elicit cell-mediated immune responses.

### 3.5. Detection of HBHA-Specific Antibodies by ELISA

HBHA/DP, LV20, and LV20/DP were injected subcutaneously on days 0,14, and 28. Forty-two days after the last immunization, the IgG, IgG1, and IgG2c against HBHA in serum were measured by ELISA. Although the titers of IgG1 and IgG2c antibodies of the LV20 group were lower than that of the HBHA/DP group, the production of IgG in mice from the LV20 group significantly increased compared with the BCG group (*p* < 0.05; Table 3). Moreover, the LV20/DP group produced significantly higher levels of IgG and IgG1 than the LV20 and BCG group (*p* < 0.05) and was similar to the HBHA/DP group (Table 3 and Figure 6A,B). These data indicated that LV20 plus DDA and Poly I:C enhanced the HBHA-specific IgG1 and IgG2c production.

## 4. Discussion

In this study, we demonstrated that the B/IC expression system is a feasible way to express and produce LV20 VLPs. We confirmed that inserting partial fragments of HBHA and MTP into the major surface glycoprotein HA of the influenza virus does not affect the stability, self-assembly, and morphology of LV20 VLPs in vitro. More importantly, LV20 VLPs can induce strong antigen-specific humoral and cell-mediated immunity targeting HBHA in mice. The B/IC expression system is a safe and effective vaccine delivery platform. Compared to traditional delivery platforms, virus-like particles have similar sizes and shapes to viruses, and increase the probability of interacting with antigen-presenting cells and B cells [33], providing natural immune advantages.

Mice receiving LV20/DP vaccine generated a higher frequency of IFN-γ- and IL-2-producing CD4^+^ and CD8^+^ T lymphocytes than HBHA/DP, BCG, and PBS groups (*p* < 0.05), which reflected that LV20/DP induced Th1-type immune response. It is well known that Th1-type immune responses play an active role in the control of M. tuberculosis infection [36]. IFN-γ induces the killing of *M. tuberculosis* in mouse macrophages [8,37]. IFN-γ deficient mice and patients showed increased susceptibility to *M. tuberculosis* infection [38,39,40], providing strong evidence that IFN-γ is required for defense against *M. tuberculosis*. IL-2 is essential for cell-mediated immunity and granuloma formation in *M. tuberculosis* infection. Studies suggest that IL-2 may provide immunotherapy to treat multidrug-resistant tuberculosis (MDR-TB). Mice receiving IL-2 combined with antibacterial agents had fewer lung lesions than mice treated with antibacterial agents alone [41].

In addition, LV20/DP vaccine-immunized mice generated higher titers of HBHA-specific IgG and IgG1 antibodies than LV20 and BCG (*p* < 0.05), indicating LV20/DP induced enhanced humoral immune responses. The antibody has been proven to enhance both innate and cell-mediated immune responses against mycobacteria [42]. Pili-mediated attachment to phagocytic cells has an important role in host defense mechanisms either through direct interaction with macrophages or by opsonization of anti-pili antibodies [43]. HBHA is associated with the extrapulmonary spread of *M. tuberculosis*. After intranasal infection of mice with wild-type mycobacteria coated with an anti-HBHA monoclonal antibody, the colonization of the *M. tuberculosis* strains in mouse spleen was impaired [20]. In addition, HBHA can mediate bacterial–bacterial interactions and mycobacterial aggregation, which can be inhibited in the presence of anti-HBHA-specific antibodies [35]. All these experiments supported the hypothesis that LV20-mediated antibodies might play a role in immune protection against Mycobacteria infection.

Our result showed that LV20/DP elicited higher HBHA-specific humoral and cellular immune responses than BCG vaccination. However, the protective efficacy against *M. tuberculosis* H37Ra induced by BCG was similar to that induced by LV20/DP (Appendix A). The reason for this might be that BCG contains more complex antigens compared to LV20.

Virus-like particles are an effective platform for antigen presentation. VLPs ranging from 20 to 200 nm can be taken up by dendritic cells (DCs) to promote the activation of T cells and B cells. DCs are a bridge between innate and adaptive immunity. DCs interact with VLPs by recognizing the same PRRs of natural viruses, such as TLRs and C-type lectin receptors (CLRs) [44,45]. The recognition and uptake of VLPs by DC cells promote the maturation of DCs, thereby stimulating of production of pro-inflammatory factors such as TNF- α and IL-1 β [46]. Pro-inflammatory factors recruit more antigen-presenting cells and increase the process of lysosomal proteolysis in DCs. It leads to the processing of VLP-based vaccines presented as MHC-peptide complexes on the surface of DCs as small peptides [47,48,49]. In some circumstances, B cells can be directly activated by VLPs without the assistance of T cells [49,50,51]. In addition, VLP-based vaccines have the advantage of activating the MHC class I-CD8^+^ T cell pathway through cross-presentation to stimulate CD8^+^cytotoxic T cells. The VLPs combine the B cell epitope, CD8^+^cell epitope, and CD4^+^cell epitope of Toxoplasma gondii, which can effectively stimulate IgG antibodies and the cellular immunity levels IFN-γ [52]. Currently, the research on virus-like particles that display Mycobacterium tuberculosis antigens is limited. Recombinant virus-like particles containing the ESAT-6 epitope induced high titers of anti-ESAT-6 serum antibodies in mice, demonstrating the viability of influenza A virus-like virus particles as platforms for the presentation of exogenous antigens [53]. HBC-VLPs containing the *M. tuberculosis* antigen CFP-10 induced higher numbers of IFN-γ secreting cells compared to splenocytes from mice immunized with CFP-10 alone or with the protein mixture [54]. However, these VLPs have small fragments of single antigens, which limit their protective efficacy and application as vaccine candidates. In this study, the influenza virus vector and baculovirus expression were used to construct the influenza virus VLPs containing foreign fragments of HBHA and MTP with a molecular weight of 20 kD, and the LV20 VLPs can effectively stimulate IgG, IgG1, and IgG2c antibodies and the cellular immunity levels IFN-γ and IL-2.

Although VLPs have a unique molecular structure that can spontaneously stimulate the immune system, the use of appropriate adjuvants can increase the immunogenicity of the vaccines based on VLPs and alter the type of immune response [55]. Aluminum salts are the most widely used adjuvants in vaccines and can be used in all approved VLPs vaccines [28]. VLP-based vaccines with aluminum salt adjuvant, such as Cervarix (HPV vaccine), Gardasil (HPV vaccine), and Hecolin (HEV vaccine), have been applied to commercialization [56]. In addition, various adjuvants such as liposomes, pathogen pattern recognition receptor (PRR) agonists, and cytokines have been used for VLP vaccines [57]. Poly I: C is a TLR3 agonist; a study has shown that using filovirus VLPs with Poly I: C can effectively enhance long-term protection against the Ebola virus [58]. Cationic liposomal DDA is used for antigen delivery, which can assemble into closed vesicle bilayers in an aqueous environment. The positively charged surface of DDA readily attracts protein and DNA molecules that contain multiple negatively charged surface molecules. Furthermore, it can bind negatively charged APC cells and bring the antigens into APC cells [59]. DDA delivers endocytic antigens directly to the cytoplasm by fusion with endosomal membranes or cross-presentation and then transports the generated peptides to the endoplasmic; therefore, it could target CD8^+^ T cells by MHC class I molecules [59,60]. DDA combined with TLR3 agonist promotes antigen cross-presentation on MHC class I molecules and increases the number of antigen-specific CD8^+^ T cells producing IFN-γ and IL-2 [61,62]. In our previous research, we combined DDA liposome with Poly I:C to construct a novel adjuvant DP. The fusion protein Ag85B-MPT64_190–198_-Mtb8.4 with DP provides high protective efficacy against an M. tuberculosis H37Rv challenge [63]. In this experiment, DP adjuvant helped LV20 to generate strong antigen-specific humoral and cell-mediated immunity.

Furthermore, we preliminarily evaluated the protective efficacy of LV20 using avirulent *M. tuberculosis* strain H37Ra. The results showed that LV20/DP reduced the bacterial count by approximately 1.3 log10 CFU than PBS control and similar to the BCG group (*p* < 0.01), while LV20 alone did not provide protection (Appendix A). It might reflect the immune responses induced by LV20 to some degree. However, H37Ra is an attenuated strain whose phoP gene was mutated [64,65], and there are 53 insertions and 21 deletions compared to the virulent strain H37Rv [66]. Further evaluation using the virulent *M. tuberculosis* strain is needed in the future.

## 5. Conclusions

The fusion protein HBHA-MTP with a molecular weight of 20 kD (L20) was inserted into the influenza virus HA fragment and expressed along with matrix protein M1 of the influenza virus to generate the influenza virus-like particles LV20. Animal experiments showed that LV20 in adjuvant of DP induced high HBHA-specific humoral and cellular immune responses. It suggests that influenza virus particles could be used to display *M. tuberculosis* antigens to design novel TB vaccines.

## 6. Patents

The authors have applied for patents on LV20. There are no conflicts of interest among the authors.

## Figures and Tables

**Figure 1 vaccines-11-00941-f001:**
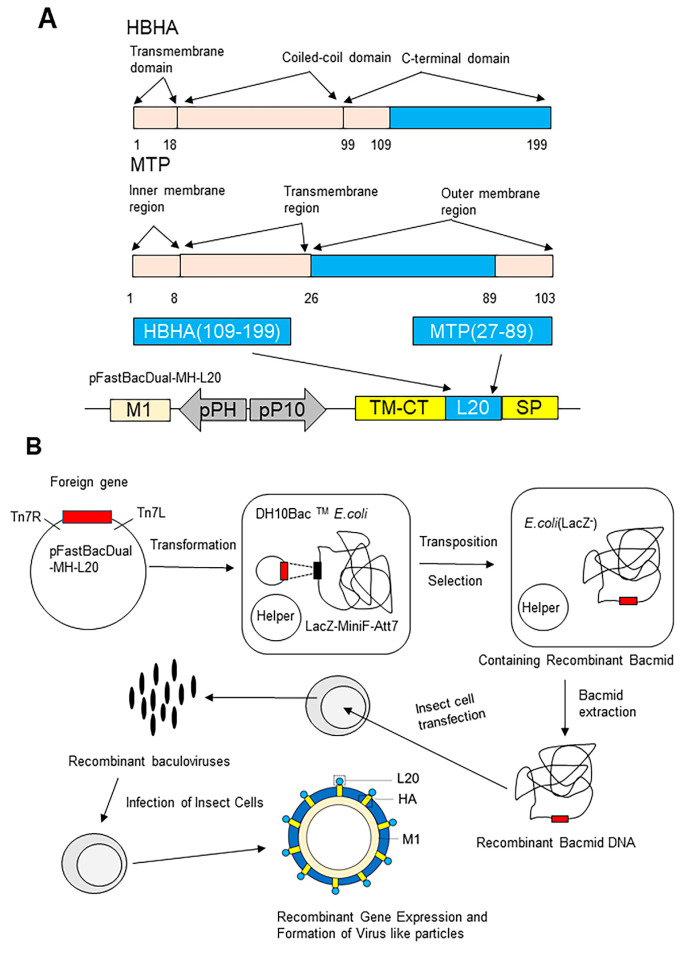
The construction and formation of LV20. (**A**) The construct of pFastBacDual-MH-L20. The functional domains found in HBHA can be divided into transmembrane domain (1–18), coiled-coil domain (19–99), and C-terminal domain (100–199). The functional domains found in MTP can be divided into inner membrane region (1–8), transmembrane region (9–26), and outer membrane region (27–103). The linked outer membrane region of HBHA and MTP (L20 in blue) were inserted between the transmembrane region and extracellular signal peptide of HA (represented in yellow) to construct the recombinant double expression plasmid pFastBacDual-MH-L20. M1, influenza matrix protein; pP10, P10 promoter; SP, extracellular signal peptide of HA; and TM-CT, transmembrane-cytoplasmic tail of HA. (**B**) The recombinant gene expression and formation of virus-like particles. The recombinant double expression plasmid pFastBacDual-HM-L20 was transformed into DH10 Bac™ E. coli (Invitrogen, Carlsbad, CA, USA) strain, and positive clones contained the recombinant bacmid -HM-L20 were obtained by antibiotic and blue–white screening. The correct recombinant bacmid DNA was extracted and transfected into Sf9 cells to obtain recombinant baculovirus. Virus-like particles were produced by recombinant baculovirus expression system.

**Figure 2 vaccines-11-00941-f002:**
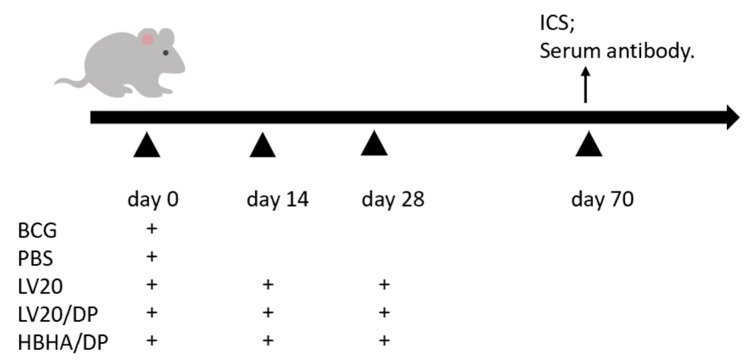
The immunization schedule to assess immune responses. C57BL/6 mice were immunized with BCG and PBS one-time inoculation on day 0. HBHA/DP, LV20, and LV20/DP were injected subcutaneously three times on days 0, 14, and 28 s. Flow cytometry and intracellular cytokine staining (ICS) and serum antibody profiles were evaluated on day 70. The number of mice per group was at least 3.

**Figure 3 vaccines-11-00941-f003:**
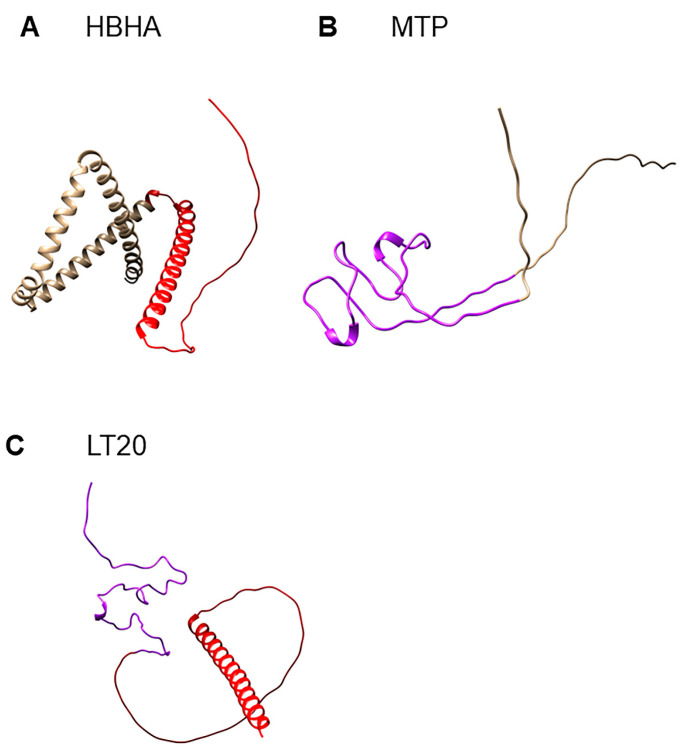
The structures of HBHA, MTP, and fusion protein L20. The protein structures were predicted by AlphaFold2. (**A**) The selected sequence of HBHA for L20 was represented in red color. (**B**) The selected sequence of MTP for L20 was represented in purple color. (**C**) L20 containing the selected sequences of HBHA and MTP.

**Figure 4 vaccines-11-00941-f004:**
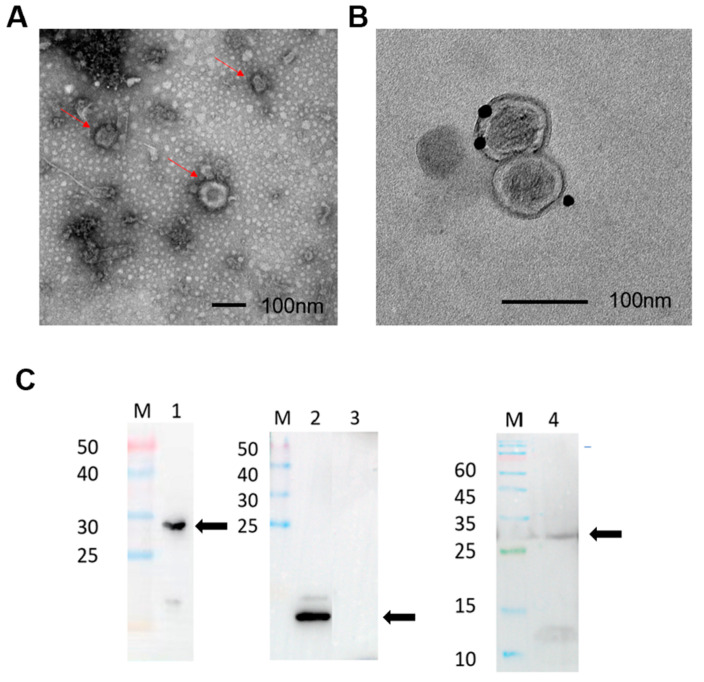
Formation and identification of LV20. (**A**) TEM pictures of prepared LV20 VLPs. Scale bar = 100 nm. (**B**) Transmission electron micrographs of immunogold-labeled LV20 VLPs. Anti-HBHA polyclonal antibody as primary antibody. Scale bar = 100 nm. (**C**) Western blotting identification of LV20 VLPs. Lane 1, identification of M1 protein with anti-M1 rabbit monoclonal antibody. Lane 2, positive control (MTP); Lane 3, negative control (LT28); Lane 4, identification of LV20 protein with anti-MTP mouse polyclonal antibody. The locations of target protein sites are marked with black arrows.

**Figure 5 vaccines-11-00941-f005:**
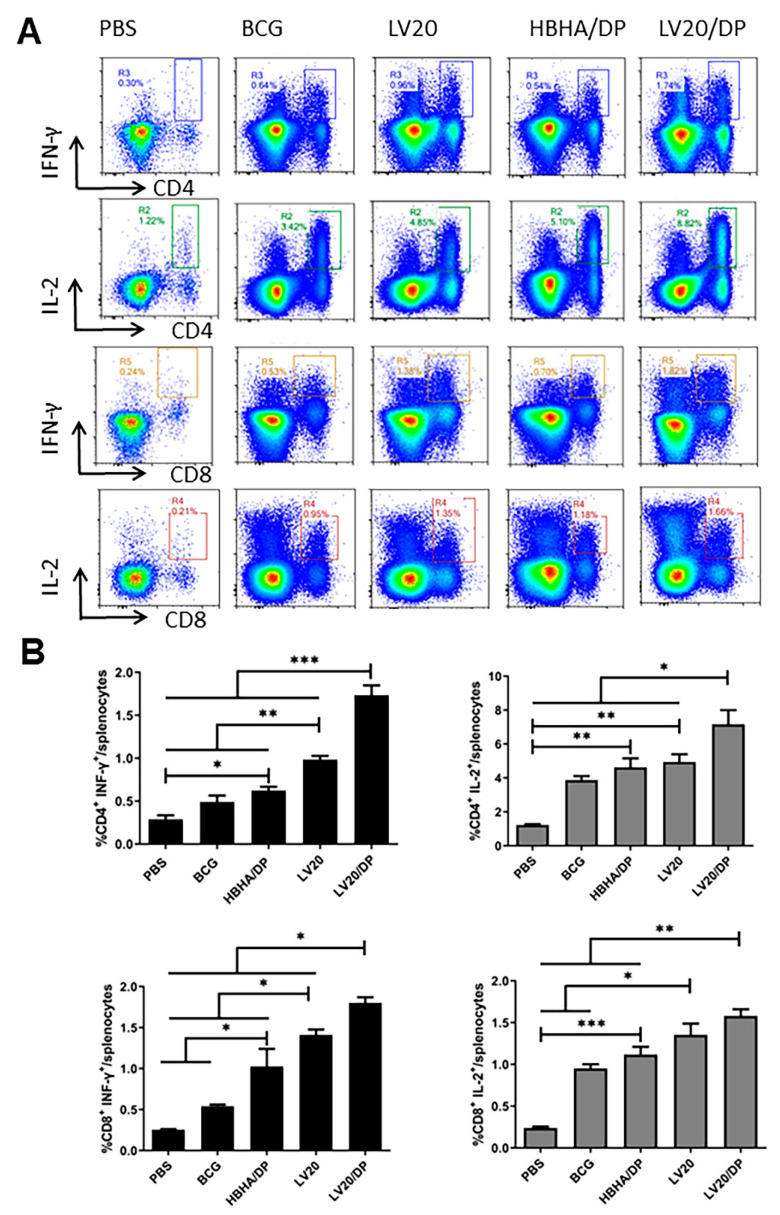
Frequencies of CD4^+^ and CD8^+^ T cells producing antigen-specific cytokines in mice. Forty-two days after the last immunization, the mouse spleen lymphocytes were isolated and stimulated with antigen HBHA in vitro. Intracellular production of cytokines was analyzed using flow cytometry. (**A**) The frequency of CD4^+^/CD8^+^ T cells secreting IFN-γ and IL-2 was analyzed by flow cytometry. (**B**) Statistical analysis of the proportion of IFN-γ- and IL-2-producing CD4^+^/CD8^+^ T cells. At least three mice per group. Mean ± SD, * *p* <0.05; ** *p* < 0.01; and *** *p* < 0.001.

**Figure 6 vaccines-11-00941-f006:**
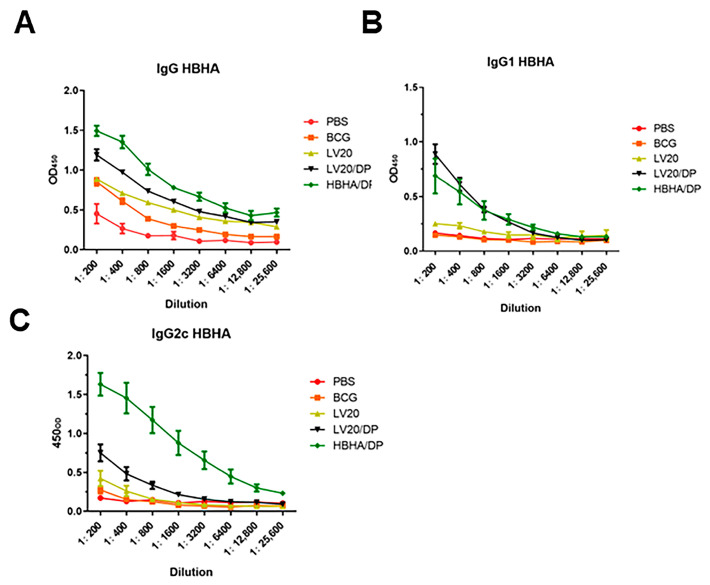
Detection of HBHA-specific antibodies by ELISA. The schedule of immunization experiment was performed according to Figure 2. Forty-two days after last immunization, the IgG, IgG1, and IgG2c against HBHA in serum were measured by ELISA. (**A**) The level of the HBHA-specific IgG. (**B**) The level of the HBHA-specific IgG1. (**C**) The level of the HBHA-specific IgG2c. Data were expressed as means ± SD. At least three mice per group.

**Table 1 vaccines-11-00941-t001:** Sequences of HBHA and MTP selected for fusion protein L20.

No.	Antigen	Position	Peptide Sequence
1	HBHA	109–199	LRSQQSFEEVSARAEGYVDQAVELTQEALGTVASQTRAVGERAAKLVGIELPKKAAPAKKAAPAKKAAPAKKAAAKKAPAKKAAAKKVTQK
2	MTP	27–89	AQSAAQTAPVPDYYWCPGQPFDPAWGPNWDPYTCHDDFHRDSDGPDHSRDYPGPILEGPVLDDPGAAPPPPAAGGGA

Notes. The amino acid sequence of HBHA was selected from 109 to 199, and the amino acid sequence of MTP was selected from 27 to 89.

**Table 2 vaccines-11-00941-t002:** Epitopes of HBHA and MTP predicted by the Immune Epitope Database (IEDB).

Type of Peptides	Location	Peptide Sequence
CD4+ T cell epitopes	HBHA_(40–54)_	EETRTDTRSRVEESR
	HBHA_(181–195)_	AAAKKAPAKKAAAKK
	HBHA_(175–189)_	AAPAKKAAAKKAPAK
	HBHA_(163–177)_	AAPAKKAAPAKKAAP
	MTP_(27–41)_	AQSAAQTAPVPDYYW
	MTP_(63–77)_	DFHRDSDGPDHSRDY
	MTP_(66–80)_	RDSDGPDHSRDYPGP
	MTP_(88–102)_	DDPGAAPPPPAAGGG
CD8+ T cell epitopes	HBHA_(37–45)_	ERAEETRTD
	HBHA_(55–63)_	ARLTKLQED
	HBHA_(113–121)_	QSFEEVSAR
	HBHA_(191–199)_	AAAKKVTQK
	HBHA_(153–161)_	KLVGIELPK
	MTP_(32–41)_	QTAPVPDYYW
	MTP_(39–47)_	YYWCPGQPF
	MTP_(94–102)_	PPPPAAGGG
B cell epitopes	HBHA_(62–79)_	EDLPEQLTELREKFTAEE
	HBHA_(107–121)_	ERLRSQQSFEEVSAR
	HBHA_(124–135)_	GYVDQAVELTQE
	HBHA_(146–153)_	AVGERAAK
	HBHA_(160–195)_	PKKAAPAKKAAPAKKAAPA KKAAAKKAPAKKAAAKK
	MTP_(30–39)_	AAQTAPVPDY
	MTP_(55–80)_	WDPYTCHDDFHRDSDGPD HSRDYPGP
	MTP_(84–100)_	GPVLDDPGAAPPPPAAG

Notes. T cell and B cell epitopes of HBHA and MTP were predicted by IEDB; the data here suggest that selected sequences were rich in T and B cell epitopes.

**Table 3 vaccines-11-00941-t003:** Serum antibodies against HBHA in mice immunized with BCG, LV20, and HBHA.

Groups	IgG	IgG1	IgG2c	IgG2c/IgG1
PBS	-	-	-	-
BCG	2.90 ± 0	2.30 ± 0	2.30 ± 0	1.00
HBHA/DP	4.21 ± 0.17 *	3.20 ± 0.30 *	4.21 ± 0.35 *	1.32
LV20	3.71 ± 0.17 *#	2.60 ± 0 #	2.60 ± 0.30 #	1.00
LV20/DP	4.01 ± 0.17 *	3.20 ± 0 *	3.00 ± 0.17 *#	0.94

Note: Forty-two days after the last immunization and the titers of anti-HBHA antibody were detected by ELISA. At least three mice per group. Mean ± SD, * *p* < 0.05 vs. BCG group; # *p* < 0.05 vs. HBHA/DP group.

## Data Availability

All data are contained within the article.

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
