# Peer review of "A VLP-Based Vaccine Displaying HBHA and MTP Antigens of *Mycobacterium tuberculosis* Induces Protective Immune Responses in *M. tuberculosis* H37Ra Infected Mice"

_vaccines, 2023, doi:10.3390/vaccines11050941_

Round 1

Reviewer 1 Report

The manuscript entitled "A VLP-based Vaccine Displaying HBHA and MTP Antigens of Mycobacterium tuberculosis Induce Strong Protective Immune Responses Against Mycobacteria" reports findings on the development of a VLP-based tuberculosis vaccine, the immune responses and protectivity in mice. The concerns below are required to be addressed:

- In the title, "... against mycobacteria" is not true because mycobacteria are a large group of Mycobacterium spp. The study included only M. tuberculosis H37Ra.

- Line 23: M1 protein was also included in the VLP.

- Line 29-30: The immune responses were antigen-specific. 

- Line 33: Insect cell expression system is not a vaccine delivery system. It is a protein production system. 

- Line 111 and 284: "Bioinformatic" instead of "Bioinformation".

- Figure 1 was provided after Figure 3.

- Why were the Tables 1, 2 and Figure 2 given in the methods part? Were they not the results of bioinformatic analysis?

- Fig2C: L20 or LT20?

- Fig3A: Show the location of Histag.

- Line 130: "Specific pathogen-free" instead of "Special pathogen-free".

- Part 2.4.1: The cloning method should be more descriptive providing primers, restriction enzymes etc.

- Line 220: The amount of proteins included in the vaccine formulation should be provided. 

- Lines 227 and 238: "6-8 weeks after the last immunization"? In other parts it was given as six weeks.

- Line 228: Three mice for a grup of vaccination is not sufficient.

- Why were different immunization shcedules used for the evaluation of immune responses and protection? Could the results be comparable?

- Instead of weeks, the days should be given for the immunization and challenge.

- Fig4C: In Lane 1, was the M1 protein loaded or LV20? Why were there two bands in Lane 1 and 4? Why was the LV20 not 20 kDa?

- In Discussion part, it should be discussed that the immune responses were measured using antigens specific to M. tuberculosis and the vaccines were composed of these proteins only, which might be the reason for the low immune responses in BCG group but the protectivity was not higher than BCG.

Author Response

Institute of Pathogen Biology,

School of Basic Medical Sciences,

Lanzhou University,

199 West Donggang Road, Lanzhou, China

Email: bdzhu@lzu.edu.cn

                                                              April 19, 2023

Dear Editor,

Thanks for your kind consideration for our manuscript. We appreciate the helpful comments from reviewer very much. The reviewer’s comments are valuable and very helpful for improving our paper. We have carefully read all comments and revised our manuscript accordingly. Our point-by-point responses are as follows. In the revised manuscript, the revision for Reviewer # 1 comments have been blue highlighted, the revision for Reviewer # 2 comments have been green highlighted, the revision for reviewer # 3 comments have been yellow highlighted. Please let us know if there are more questions on our manuscript.

Response to Reviewer #1 Comments

Comment 1: - In the title, "... against mycobacteria" is not true because mycobacteria are a large group of Mycobacterium spp. The study included only M. tuberculosis H37Ra.

Response: Thank you for your suggestion. As suggested, we have added H37Ra in the title.

Comment 2: - Line 23: M1 protein was also included in the VLP.

Response: Thank you for your suggestion. We have revised as suggested.

Comment 3: - Line 29-30: The immune responses were antigen-specific.

Response: Thank you for your suggestion. We have revised as suggested.

Comment 4: - Line 33: Insect cell expression system is not a vaccine delivery system. It is a protein production system.

Response: Thank you for your suggestion. We have revised as suggested.

Comment 5: - Line 111 and 284: "Bioinformatic" instead of "Bioinformation".

Response: Thank you for your careful revision. As suggested, this sentence has been revised.

Comment 6: - Figure 1 was provided after Figure 3.

Response: Thank you for your careful revision. This error has been corrected.

Comment 7: - Why were the Tables 1, 2 and Figure 2 given in the methods part? Were they

not the results of bioinformatic analysis?

Response: Thank you for your perfect suggestion. They have been moved to the part of result.

Comment 8: - Fig2C: L20 or LT20?

Response: Thank you for your question, LT20 is correct. It was defined in the abstract.

Comment 9: - Fig3A: Show the location of His tag.

Response: Thank you for your question. There is no His tag built in the research.

Comment 10: - Line 130: "Specific pathogen-free" instead of "Special pathogen-free".

Response: Thank you for your careful revision. As suggested, this sentence has been revised.

Comment 11: - Part 2.4.1: The cloning method should be more descriptive providing primers, restriction enzymes etc.

Response: Thank you for your suggestion. This information has been added to the method section.

Comment 12: - Line 220: The amounts of proteins included in the vaccine formulation should be provided.

Response: Thank you for your suggestion. This information has been added to the method section.

Comment 13: - Lines 227 and 238: "6-8 weeks after the last immunization"? In other parts it was given as six weeks.

Response: Thank you for your careful revision. This error has been corrected.

Comment 14: - Line 228: Three mice for a grup of vaccination is not sufficient.

Response: Thank you for your suggestion. Three mice per group is the minimum number for the experiment as there were obvious difference were observed. In the experiment of protective efficacy, there are 5-6 mice per group.

Comment 15: - Why were different immunization schedules used for the evaluation of immune responses and protection? Could the results be comparable?

Response: Thank you for your question. As for the immune responses, a traditional schedule was applied. Meanwhile, our lab found that prolonging the intervals of immunization could improve the protective efficacy compared with traditional schedule of 3 doses at 3-week intervals. Therefore, in the evaluation of protective efficacy against M. tuberculosis H37Ra, we chose an immunization schedule of 0-4-12 weeks. We have added explanation in the manuscript.

Reference: Bai, C., et al., Prolonged intervals during Mycobacterium tuberculosis subunit vaccine boosting contributes to eliciting immunity mediated by central memory-like T cells. Tuberculosis (Edinb), 2018. 110: p. 104-111.

Comment 16: - Instead of weeks, the days should be given for the immunization and challenge.

Response: Thank you for your suggestion. We have revised as suggested.

Comment 17: - Fig4C: In Lane 1, was the M1 protein loaded or LV20? Why were there two bands in Lane 1 and 4? Why was the LV20 not 20 kDa?

Response: Thank you for your question. Lane 1 is M1 protein loaded. The pFastBacDual vector has two promoters, pP10 and pPH. The transmembrane region and extracellular signal peptide of HA and the matrix protein M1 of influenza virus were inserted downstream of the pP10 and pPH promoters, respectively. The fusion protein containing L20 and the transmembrane region and extracellular signal peptide of the influenza virus HA, which were approximately 27 kDa (Figure 4C, Lane4). The molecular weight of M1 (Figure 4C, Lane1) is 29 kDa.

Comment 18: - In Discussion part, it should be discussed that the immune responses were measured using antigens specific to M. tuberculosis and the vaccines were composed of these proteins only, which might be the reason for the low immune responses in BCG group but the protectivity was not higher than BCG.

Response: Thank you for your suggestion. As suggested, we have added this discussion in the manuscript.

We have revised our manuscript very carefully. If there are any questions, please let us know and we will do more revisions. We are looking forward to hearing from you.

Best wishes,

Juan Wang

Bingdong Zhu, M.D.

Reviewer 2 Report

The first part of the manuscript, which deals with the construction of novel recombinant vaccine candidates for TB, is significantly important and thoroughly explained. Therefore, this section definitely deserves scientific merit.

However, the usage of H37Ra, an attenuated and non-virulent strain, for mice challenge studies to evaluate the vaccine candidates, significantly reduced the innovation and importance of this study.  It is genuinely unacceptable and inappropriate to use an avirulant strain for crucial vaccine-mediated protection against infection challenge studies.  In addition, the authors have interchangeably used the word "mycobacterium tuberculosis" throughout this paper, to confuse the reader and to generalize between the "truely virulent Mtb strains" and the "non-pathogenic, avirulent strains, such as H37Ra that the authors have used in this study.  This is wrong and inappropriate. Non-availability of a BSL3 was given as an excuse for this mistake, which is very vague and questions  the scientific rigor of this study.

I  encourage the authors to read some high impact manuscripts from established groups in TB research to understand the pathogenic and virulence differences between pathogenic and non-pathogenic Mtb strains in mice model.  Therefore, I strongly recommend the authors to revise the manuscript by including the mice challenge study using pathogenic, virulent Mtb strains, preferably, using a clinical isolate, which will definitely improve the scientific validity of this study.  Alternately, the authors should consider removing the challenge study data and present only the immunogenicity data of their recombinant vaccine candidates.

Author Response

Dear Reviewer # 2,

Thank you very much for your affirmation and great suggestions for our research. We appreciate the helpful comments from reviewer very much. The reviewer’s comments are valuable and very helpful for improving our paper. In the revised manuscript, the revision for Reviewer # 1 comments have been blue highlighted, the revision for Reviewer # 2 comments have been green highlighted, the revision for reviewer # 3 comments have been yellow highlighted.

We agree with you that challenging with H37Ra, an attenuated strain, to evaluate the vaccine candidates is insufficient. This is the limitation of this study. Due to biosafety facility limitations, avirulent M. tuberculosis strain H37Ra instead of virulent strain was applied to preliminarily evaluate protective efficacy of LV20. H37Ra challenge is not as good as virulent M. tuberculosis challenge, but it has been used to preliminarily evaluate protective efficacy of TB vaccines. H37Ra and LV20 all contained the antigens HBHA and MTP, our result show LV20/DP induced higher protective efficacy against M. tuberculosis H37Ra than LV20 without adjuvant and PBS control. To some extent, it reflects the protective immune responses induced by LV20. In future, we may use M. tuberculosis challenge to further confirm the protective efficacy.

In addition, in order to avoid confusing the readers, we have revised the parts of the manuscript that may cause the misunderstand. In the title and discussion part, the words of attenuated H37Ra were added.

We have revised our manuscript very carefully. If there are any questions, please let us know and we will do more revisions. We are looking forward to hearing from you.

Best wishes,

Juan Wang

Bingdong Zhu, M.D.

Reviewer 3 Report

The authors of the article “A VLP-based Vaccine Displaying HBHA and MTP Antigens of Mycobacterium tuberculosis Induce Strong Protective Immune Responses Against Mycobacteria” described a new viral construct displaying highly immunogenic Mycobacterium tuberculosis recombinant proteins as candidate vaccine to protect against Tuberculosis. The article is well written, scientific and technical questions are well addressed and answered. The discussion is also well written. However, I would like to address some questions and minor comments to the authors:

1-    The vaccine preparation and immunization figure (Figure1) should be improved. We don’t understand how many groups you have, when are scheduled days of immunizations and with what immunogens. The paragraph from line 218 to 233 should also better explain the vaccine protocol and differences on mice groups.

2-    Line 75, “…strong functions…”, which kind of functions are expressed?

3-    Line 87, “…are a safe and valuable platform…”

4-    Line 106 to 109, you are giving a conclusion in the introduction. Not needed here. You should tell instead the scientific question related to your study.

5-    Why for the VLP construction you took only HBHA from 109-199 (and not from 99-199) and MTP from 26-89 (and not from 26-103) (fig 3-A)?

6-    Line 197, “…2.6.1 Detection of VLPs by transmission electron…”  

7-    Line 227, “…at 6-8 weeks…”, in the figure you waited 6 weeks, be clear in the paragraph.

8-    Why did you used two different immunization schedules to assess the Immune response and Protection? Why did you not used only one protocol for both? It is very complicated to compare two different immune responses.

9-    Fig 1.B, why the time between prime and boost are not the same (0, 4 and 12 weeks)?

10- Line 246, from where you isolated lymphocytes?

11- Line 277, did you weighted mice during the infection to check their sickness?

12- Figure 4-C, please add more contrast, some lines are not visible. Add an arrow () to show the LV20 protein of interest.

13- Figure 6-A, how can you detect specific HBHA IgG titers in the PBS control? These are normally pathogen-free mice.

14- Line 506, it’s not a chimeric VLP because you are displaying only one type of protein and not different kind of proteins.

Author Response

Dear Reviewer # 3,

Thanks for your kind consideration for our manuscript. We appreciate the helpful comments from reviewer very much. The reviewer’s comments are valuable and very helpful for improving our paper. We have carefully read all comments and revised our manuscript accordingly. Our point-by-point responses are as follows. In the revised manuscript, the revision for Reviewer # 1 comments have been blue highlighted, the revision for Reviewer # 2 comments have been green highlighted, the revision for reviewer # 3 comments have been yellow highlighted.

We have revised our manuscript very carefully. If there are any questions, please let us know and we will do more revisions. We are looking forward to hearing from you.

Best wishes,

Juan Wang

Bingdong Zhu, M.D.

Response to Reviewer #3 Comments

The authors of the article “A VLP-based Vaccine Displaying HBHA and MTP Antigens of Mycobacterium tuberculosis Induce Strong Protective Immune Responses Against Mycobacteria” described a new viral construct displaying highly immunogenic Mycobacterium tuberculosis recombinant proteins as candidate vaccine to protect against Tuberculosis. The article is well written, scientific and technical questions are well addressed and answered. The discussion is also well written. However, I would like to address some questions and minor comments to the authors:

Comment 1: -The vaccine preparation and immunization figure (Figure1) should be improved. We don’t understand how many groups you have, when are scheduled days of immunizations and with what immunogens. The paragraph from line 218 to 233 should also better explain the vaccine protocol and differences on mice groups.

Response: Thanks for your suggestion. The corresponding modifications about the method and figure have been made.

Comment 2: - Line 75, “…strong functions…”, which kind of functions are expressed?

Response: Thanks for your suggestion. As suggested, this sentence has been revised

Comment 3: - Line 87, “…are a safe and valuable platform…”

Response: Thank you for your careful revision. As suggested, this sentence has been revised.

Comment 4: -Line 106 to 109, you are giving a conclusion in the introduction. Not needed here. You should tell instead the scientific question related to your study.

Response: Thank you for your suggestion. As suggested, this sentence has been revised

Comment 5: -Why for the VLP construction you took only HBHA from 109-199 (and not from 99-199) and MTP from 26-89 (and not from 26-103) (fig 3-A)?

Response: Thank you for your question. Considering that large fragments may affect the solution of of fusion protein, we selected fragments of them according to the results of bioinformation analysis results.

Comment 6: -Line 197, “…2.6.1 Detection of VLPs by transmission electron…”

Response: Thank you for your suggestion. As suggested, this sentence has been revised

Comment 7: -Line 227, “…at 6-8 weeks…”, in the figure you waited 6 weeks, be clear in the paragraph.

Response: Thank you for your suggestion. As suggested, this error has been corrected.

Comment 8: - Why did you used two different immunization schedules to assess the Immune response and Protection? Why did you not used only one protocol for both? It is very complicated to compare two different immune responses.

Response: Thank you for your question. As for the immune responses, a traditional schedule was applied. Meanwhile, our lab found that prolonging the intervals of immunization could improve the protective efficacy compared with traditional schedule of 3 doses at 3-week intervals. Therefore, in the evaluation of protective efficacy against M. tuberculosis H37Ra, we chose an immunization schedule of 0-4-12 weeks.

Reference: Bai, C., et al., Prolonged intervals during Mycobacterium tuberculosis subunit vaccine boosting contributes to eliciting immunity mediated by central memory-like T cells. Tuberculosis (Edinb), 2018. 110: p. 104-111.

Comment 9: -Fig 1.B, why the time between prime and boost are not the same (0, 4 and 12 weeks)?

Response: Thank you for your question. It is believed that central memory T cells (TCM) provide long-term protection against tuberculosis (TB). Research shows an interval of 2–3 months between the prime and the boost is beneficial to increase the number and function of long-lived memory T cells and improve the protective efficacy.

Reference: Sallusto, F., et al., From vaccines to memory and back. Immunity, 2010. 33(4): p. 451-63.

Comment 10: - Line 246, from where you isolated lymphocytes?

Response: Thank you for your suggestion. As suggested, this sentence has been revised

Comment 11: - Line 277, did you weighted mice during the infection to check their sickness?

Response: Thanks for your great suggest. As we used the avirulent strain for infection, we did not detect the change. However, this is a very great suggest that we should pay closer attention. We are sorry for not detecting the body weight of the mice this time. In future, we will pay attention to it.

Comment 12: - Figure 4-C, please add more contrast, some lines are not visible. Add an arrow () to show the LV20 protein of interest.

Response: Thank you for your suggestion. As suggested, this figure has been improved.

Comment 13: -Figure 6-A, how can you detect specific HBHA IgG titers in the PBS control? These are normally pathogen-free mice.

Response: Thank for your question. When the dilution ratio of serum is too low, it may cause non-specific reactions of related antigens. This non-specific reaction is improve as the dilution ratio increases.

Comment 14: -Line 506, it’s not a chimeric VLP because you are displaying only one type of protein and not different kind of proteins.

Response: Thank you for your careful revision. As suggested, this sentence has been revised.

Round 2

Reviewer 1 Report

The authors mostly addressed the concerns of this reviewer after revision. However, still there are some points to be revised:

- Title: The expression "Mycobacteria H37Ra" is not accurate. It should be "M. tuberculosis H37Ra". Also it is better to add "in mice".

- Abstract Line 29: Again it is better to add "in mice" after "vaccination".

- Line 213: ... subcutaneously "vaccinated" with...

- Lines 220, 229, 230, 237 and so on: If you indicate the days as time points, use "at days X, XX, and XXX" not "at X, XX and XXX days". You can use it as "after XX days".   

Author Response

Dear Reviewer # 1,

Thank you for your time, patience and kindly help. We have carefully read all comments and revised our manuscript accordingly. Our point-by-point responses are as follows.

Response to Reviewer #1 Comments

Comment 1: - Title: The expression "Mycobacteria H37Ra" is not accurate. It should be "M. tuberculosis H37Ra". Also it is better to add "in mice".

Response: Thank you for your suggestion. According to the suggestion of Reviewer#2, we have moved the experiment of the H37Ra challenge to the supplementary section, and combined the suggestion of you the title has been revised as “A VLP-based Vaccine Displaying HBHA and MTP Antigens of Mycobacterium tuberculosis Induces Strong Antigen- Specific Immune Responses in Mice”. Thank you for your understanding.

Comment 2: - Abstract Line 29: Again it is better to add "in mice" after "vaccination".

Response: Thank you for your great suggestion. We have added "in mice" after "vaccination" in the manuscript.

Comment 3: - Line 213: ... subcutaneously "vaccinated" with...

Response: Thank you for your great suggestion. We have revised as suggested.

Comment 4: - Lines 220, 229, 230, 237 and so on: If you indicate the days as time points, use "at days X, XX, and XXX" not "at X, XX and XXX days". You can use it as "after XX days".  

Response: Thank you for your great suggestion. We have carefully revised them in the manuscript.

If there are any questions, please let us know and we will do more revisions. We are looking forward to hearing from you.

Best wishes,

Juan Wang

Bingdong Zhu, M.D.

Reviewer 2 Report

I went through the authors' responses to my comments in this manuscript (ID: vaccines-2331684) and I have serious concerns about the suitability of this article for publication. The major issue is that the vaccine efficacy testing in mice was not conducted with a pathogenic Mycobacterium tuberculosis bacterium, instead, the authors have used a non-pathogenic, avirulent strain.

This is a crucial experiment/study in this paper, which is to test the efficacy of a test vaccine against TB in an animal model. Therefore, it must be performed with a pathogenic/virulent Mtb strain. The author's response to this question is not scientifically justified and is unacceptable in the field. 

Author Response

Response to Reviewer #2 Comments

Response:

Thank you for your time, patience and kindly help. We agree with you that challenging with virulent M. tuberculosis H37Rv to evaluate the TB vaccine candidates is more clinically meaningful than avirulent H37Ra. However, due to the limitations of ABSL3 and COVID-19 epidemic we preliminarily evaluated the protective efficacy using H37Ra strain, which might reflect the immune responses induced by LV20 to some degree. According to the suggestion, we have moved the content on challenge experiment to the supplementary section and revised the relevant parts in the manuscript. If necessary, we can remove it from the manuscript.

H37Ra is a stable attenuated strain obtained from H37Rv. Comparative genomic studies have shown that compared to the H37Rv genome, there are 53 insertions and 21 deletions in H37Ra. The mutations in the phoP gene may also be one of the reasons for the decay in H37Ra. 

Reference:

  1. Zheng, H., et al., Genetic basis of virulence attenuation revealed by comparative genomic analysis of Mycobacterium tuberculosis strain H37Ra versus H37Rv. PLoS One, 2008. 3(6): p. e2375.
  2. Ryndak, M., S. Wang, and I. Smith, PhoP, a key player in Mycobacterium tuberculosis virulence. Trends Microbiol, 2008. 16(11): p. 528-34.
  3. Broset, E., C. Martin, and J. Gonzalo-Asensio, Evolutionary landscape of the Mycobacterium tuberculosis complex from the viewpoint of PhoPR: implications for virulence regulation and application to vaccine development. mBio, 2015. 6(5): p. e01289-15.
  4. Aagaard, C., et al., A multistage tuberculosis vaccine that confers efficient protection before and after exposure. Nat Med, 2011. 17(2): p. 189-94.
  5. Giri, P.K., I. Verma, and G.K. Khuller, Protective efficacy of intranasal vaccination with Mycobacterium bovis BCG against airway Mycobacterium tuberculosis challenge in mice. J Infect, 2006. 53(5): p. 350-6
  6. Deng, Y.H., H.Y. He, and B.S. Zhang, Evaluation of protective efficacy conferred by a recombinant Mycobacterium bovis BCG expressing a fusion protein of Ag85A-ESAT-6. J Microbiol Immunol Infect, 2014. 47(1): p. 48-56.

If there are any questions, please let us know and we will do more revisions. We are looking forward to hearing from you.

Best wishes,

Juan Wang

Bingdong Zhu, M.D.

Round 3

Reviewer 2 Report

I suggest the authors to keep their original title with slight modification as follows:  "A VLP-based Vaccine Displaying HBHA and MTP Antigens of Mycobacterium tuberculosis Induce Protective Immune Responses in Mycobacterium tuberculosis H37Ra Infected Mice"

Author Response

Dear Reviewer #2

Thank you for your time, patience and kindly help. We have carefully read the comment and revised our manuscript accordingly.

Response to Reviewer #2 Comments

Comment: I suggest the authors to keep their original title with slight modification as follows:  "A VLP-based Vaccine Displaying HBHA and MTP Antigens of Mycobacterium tuberculosis Induce Protective Immune Responses in Mycobacterium tuberculosis H37Ra Infected Mice"

Response: Thank you for your great suggestion. I have revised the title and resubmitted the manuscript.

If there are any questions, please let us know and we will do more revisions. We are looking forward to hearing from you.

Best wishes,

Juan Wang

Bingdong Zhu, M.D.